# A Prototype of an Electromagnetic Induction Sensor for Non-Destructive Estimation of the Presence of Corrosive Chemicals Ensuing Concrete Corrosion

**DOI:** 10.3390/s19091959

**Published:** 2019-04-26

**Authors:** Kabir A. Mamun, Ravin N. Deo, F. R. Islam, Hemanshu R. Pota, Aneesh A. Chand, Kushal A. Prasad, Aisake Cakacaka

**Affiliations:** 1School of Engineering and Physics, The University of the South Pacific, Suva, Fiji; aneeshamitesh@gmail.com (A.A.C.); kushalaniketp@gmail.com (K.A.P.); abcaka088@gmail.com (A.C.); 2Department of Civil Engineering, Monash University, Clayton Campus, Victoria 3800, Australia; ravin.deo@monash.edu; 3School of Science and Engineering, University of Sunshine Coast, Queensland 4556, Australia; fislam@usc.edu.au; 4School of Engineering and Information Technology, The University of New South Wales, Canberra, NSW 2612, Australia; h.pota@adfa.edu.au

**Keywords:** EMI sensor, concrete corrosion, non-destructive evaluation, single-loop coil (SLC), multiple-loop coil (MLC)

## Abstract

The corrosion of steel reinforcement in concrete often leads to huge unbudgeted expenses for maintaining, monitoring and renovating an infrastructure. This is mainly due to the presence of salts or chemical chlorides that pose a danger to the concrete structures. The determination of the existence of these corrosive salts is vital for defining the service life of concrete. This research looked at developing an electromagnetic induction (EMI) sensor for the detection of corrosive salts. The first design adopted a single-loop coil (SLC) concept, and the second was based on a multiple-loop coil (MLC) one using copper wire. Tests were conducted on these two techniques, and with the results obtained, the latter seemed more promising; thus, a prototype sensor was developed using the MLC concept. As this new prototype sensor was able to detect the manifestation of chemical contents in a concrete structure, it could be used as a non-destructive evaluation (NDE) technique for the detection of corrosive chemicals in concrete and has the further possibility of detecting corrosion in concrete.

## 1. Introduction

Corrosion is a destructive attack on a material’s surface caused by chemical or electrochemical reactions with the environment [1,2,3]. It is primarily induced by acids, sulphate, ammonium ions, magnesium ions, pure water, or alkali aggregate reactions. Although the surface deposits caused by corrosion can be beneficial, they are more often damaging to the host material and surrounding material(s) or structure(s). The corrosion of steel reinforcement is the main cause of damage to, and early failure of, reinforced concrete structures in civil engineering. It often leads to enormous unbudgeted direct costs for the inspection, maintenance, restoration or replacement of the associated infrastructure, plus, potentially, indirect costs. While the degradation of concrete can occur in several ways [1,2,3], the corrosion-induced deterioration of its reinforcing steel is the dominant cause [4,5,6,7]. 

Initiation and propagation, which are the two main stages involved in the corrosion process, are supported by various mechanisms, (i.e., diffusion, sorption, permeation and migration) [8]. During the initiation stage, aggressive (corrosive) agents are transported through the concrete layer (protective cover) [9,10], and then the subsequent propagation stage causes depassivation of the steel. In turn, this leads to an electrochemical oxidation reaction which results in the weakening (thinning) of the reinforcing steel and the spalling of the concrete [11]. The reported research focuses on one particular type of corrosion that results from chemical or electrochemical reactions to the environment: specifically, pitting [12]. As the new sensor proposed in this paper is capable of detecting the chemical contents present in a concrete structure that cause pitting, it can be used as a non-destructive evaluation (NDE) technique for the detection and further evaluation of concrete corrosion. The severity of pitting increases logarithmically with increasing amounts of chloride concentration present in a pore solution [13]. Concrete spalling and rebar (reinforcing bar) corrosion normally occur due to chemical or electrochemical reactions [14], which mainly affect the rebars embedded in the concrete. Normally, reinforced concrete is protected by the passive alkalinity of its cement matrix but, due to the ingress of aggressive environmental influences, steel reinforcements can corrode [3,15]. Therefore, it is important to study the rate at which water and chlorides penetrate a concrete structure’s protective cover layer to determine the remaining service life of an aging concrete infrastructure [11].

To detect corrosion in a concrete structure, many solutions have been developed, including NDE techniques that are normally preferred [16,17] because of their nature and cost effectiveness. Commonly used NDE techniques drawn from the literature and produced in a comprehensive guide by Ryan et al. [18] are acoustic emission, which requires the material to be under stress to detect flaws but is less effective for certain loading scenarios and sensitive to external noise [18,19]; the electrical method (half-cell method), which is only suitable for measuring the probability of the corrosion of a rebar occurring [18,20]; delamination detection machinery, which is less effective for irregular shapes and very thin materials [18,21]; the ground-penetrating radar (GPR) and magnetic method, which becomes less effective as the thickness of the concrete’s cover increases [18,22]; electromagnetic methods, such as the high-speed electromagnetic roadway mapping and evaluation system (HERMES), which have similar limitations to GPR [18,23]; the pulse velocity method, which is less effective for brittle materials and also has the same limitations as delamination detection machinery [18,24]; impact-echo testing, which requires multiple impact locations for high accuracy and is also incapable of assessing the concrete–steel bond strength [18,25]; the infrared thermography technique, which is sensitive to contaminants [18,26,27]; ultrasonic testing, which has the same limitations as previously mentioned for the pulse velocity method [18,28]; a neutron probe, a complicated and expensive approach requiring specific expertise [29]; the nuclear method, which also requires particular skills and a complex analysis [18,30]; a pachometer, which is unable to detect rebar deterioration [18]; the rebound and penetration method, which requires prior (benchmark) data for comparison with properly determined concrete strengths [18,29]; and smart concrete, which is still in the research phase [18,31].

Essentially, an appropriately configured electromagnetic induction (EMI)-based technique is capable of detecting moisture, determining the presence of cracks and estimating their sizes, determining the susceptibility of concrete to corrosion, identifying regions susceptible to chloride penetration, and mapping corrosion activity [18,32,33]. Various researchers have presented electromagnetic NDE techniques based on microwave [34,35,36], GPR [37,38], and capacitive [37,38] principles. However, in this research, the research team propose and develop a novel NDE technique suitable for monitoring or sensing the presence of corrosive chemicals responsible for concrete corrosion based on EMI principles [17]. Its focus is on detecting the types and amounts of corrosive chemical compounds present in concrete, on which predictions of the adverse effects of corrosion can be based. As the 3.5% of sodium chloride (NaCl) present in sea water is one of the main causes of pitting corrosion, it is imperative that a NaCl solution is used as a corrosive chemical for testing in this research. Although different concepts using EMI principles have been developed to detect minerals beneath the Earth’s surface [39,40,41,42,43,44,45], to date, there is no literature on an EMI-based NDE sensor being used to monitor concrete corrosion. The following Sections describe the EMI sensor principles used to design an NDE sensor for detecting corrosion in concrete, the experimental test setup and procedure, and the results obtained. 

## 2. EMI Sensor Principles 

Electromagnetic theory is based on a set of equations produced by Maxwell in 1865 that sum all the known results in the field of electricity and magnetism, with the basic law of electricity and magnetism. In Equation (1), $\nabla .\vec{E}$ is the change in electric field and $\partial \vec{B}$ is the change in magnetic field with respect to change in time.
(1)$$\nabla .\vec{E}=\frac{\partial \vec{B}}{\partial t}$$


In particular; Faraday’s law of EMI, which is the theoretical basis of many applications, is used in this research. The basic principle of the EMI method is followed, and its concept applied for the design of a concrete corrosion sensor. Normally, in EMI, a strong direct current is passed through a transmitter loop and a receiver coil is placed in its centre which produced induced voltage signal.

The currents in conductive layers decay more slowly than those in resistive ones, with the relationship among the time-varying decay amplitude, time-layer conductivity and depth quite complex. Generally, longer decay times relate to deeper depths [41,46]. 

In this research, concrete is used as the only conductor between the transmitting and receiving loops to detect the chemical properties inside the concrete. The voltage response can be separated into three different stages of early (the response is constant over time), intermediate (the shape of the response varies over time) and late (the response is a straight line on a log-log plot), and varies according to the time and conductivity as where *k_1_* = a constant, *M* = the product of the current (amps) and area (m^2^), *σ* = the terrain conductivity (Siemens/m), *t* = the time (s), and *V(t)* = the output voltage from a single turn of the receiver coil in a 1 m^2^ area.
(2)$$V(t)=\frac{k_{1}M\sigma^{3/2}}{t^{5/2}}$$


As the measured voltage changes linearly with resistivity, the EMI is intrinsically more sensitive to small variations in conductivity than conventional resistivity determinations. The measured voltage can be represented with respect to the apparent resistivity, and, since *ρ* = 1/*σ*, the resistivity with depth can be calculated by
(3)$$\rho_{a}(t)=\frac{k_{2}M^{2/3}}{e{\left(t\right)}^{2/3}t^{5/2}}.$$


The voltage induced in the receiver coil is the product of the receiver coil moment (*M_r_*) (the area multiplied by the number of turns) multiplied by the time derivative of the vertical magnetic flux density as where *µ* is the magnetic permeability and *M_t_* the transmitter loop moment (L_2_ × I). As *V*(*t*)/*M_r_* is inversely proportional to the time over which the current diffuses downwards, it is difficult to measure at great depths unless the transmitter moment is increased.
(4)$$\frac{V(t)}{M_{r}}=\frac{\mu M_{t}}{5t}{\left(\frac{\mu }{4\pi t\rho_{a}}\right)}^{3/2}$$


## 3. Sensor Design

In the design of the sensor, two aspects considered were the dimensions required and choice of material. One approach designed a SLC consisting of one loop of metallic wire with a rectangular or circular form, and the other an MLC consisting of copper wires wound in a circular form with several loops.

### 3.1. SLC

This concept was based on the sensors used in mineral prospecting [42], with the probe designs produced with two different metals, (i.e., brass and mild steel), as shown in Figure 1 (measurements in mm). They were constructed at low levels of accuracy to firstly obtain a signal by supplying current to the outer coil of the sensor and receiving a voltage signal at the inner one. Then, the accuracy of the design was considered. 

During the early stage of this research, various SLC search coils were tested varying different materials like steel, brass and copper with different diameters, coil lengths, and placements of inner and outer search coils in different configurations. To obtain greater accuracy, the lengths and thicknesses of the two coils; the gaps between the receiver and transmitter; and other factors, such as the shapes and material variations of the coils, were taken into account, and this is also evident through Equations (2) and (3). However, before focusing on constructing the sensor with improved dimensional accuracy, the various SLCs were tested and observed, and imperial data indicated that SLCs produce very low-level output signals (voltage signals); hence, an MLC design was taken into account. 

### 3.2. MLC

During the second stage of this research, a concentric MLC sensor was designed to counter the shortcomings identified in the SLC sensor design. The proposed sensor had the added advantage of providing a larger area of detection field and greater detection depth. It was basically constructed from Category 5 (CAT 5) cables, which can withstand frequencies of up to 100 MHz. A single sheathed strand extracted from the twisted-pair cable was then wound up with a coil winder. The outer coil (transmitter) and inner coil (receiver) were constructed using 12 and six turns, respectively.

Before proceeding with the MLC design, it was necessary to carry out an analysis of the behaviour of the electromagnetic field (EMF) in the region of interest. The non-commercial finite element method magnetics (FEMM) software was used to perform the 2D simulations as shown in Figure 2. The transmitter and receiver search coil was placed in concentric alignment coplanarly with a 10 mm air gap, and the sensor acted as inductively coupled (magnetic fields fluctuated around the transmitter coil, and as a result the secondary coil began to induce current). 

## 4. Experimental Design and Tests

### 4.1. Materials 

Three concrete blocks of the same mixture were tested. Standard Portland cement CEM I 42.5R (Pacific Cement Limited, Lami, Suva, Fiji) and a quick mix (with a blend ratio of 1:1 from coarse to fine aggregates, with the maximum aggregate size 20 mm) were used to cast the block specimens, which were cured for 30 days at an air temperature of 18 °C and 60% relative humidity. A mortar box frame was machined into specific sizes of a frame with an inner dimension of 150 mm × 450 mm × 450 mm, as shown in Figure 3. One block was soaked in distilled water and another in a NaCl (0.05 molar mass) solution for 24 hours before tests were conducted, while the third was left dry in ideal conditions, i.e., 25 °C in a closed room. The 24-hour time period for soaking the blocks was chosen, as the concrete blocks were saturated within 24 hours. This was evident, as concrete blocks were weighed after 12, 24, 36 and 48 hours, and it was observed that the weight did not change after the 24-hour period.

### 4.2. Testing Methods

Firstly, experimental tests were conducted on the SLC sensor using the coplanar setup shown in Figure 4a, and the design of the MLC sensor was not introduced until these tests produced unsatisfying and inconclusive results. 

The MLC design was similar to that of the metal detection technique used by Nabighian [41] but applied in a new way to detect corrosion in rebars. The outer search coil was the transmitting port for the current supply and worked as a transmitter, while the inner coil acted as a receiver that produced voltage signals, as shown in Figure 4b. The frequency of the input current was 20 kHz, and the voltage was supplied to the transmitter in square pulses. The signal obtained by the receiver had a sudden spike and then a gradual decline, as anticipated, and was similar to electromagnetic receiver signals [41]. The experimental setup is shown in Figure 5a. During the first set of experiments, the sensor was placed coplanarly on the concrete block’s top surface. The ISO-TECH function generator was used to supply current to the transmitter coil, (i.e., outer loops) at a frequency of 30 kHz. The NI cDAQ-9174 (National Instruments, US), which contained the NI-9215 current input module, can handle frequencies of up to 100 kHz. Finally, the voltage induced on the inner coils was measured using a LabVIEW routine. 

The same setup and procedures for generating and obtaining signals as in the first set of tests for the blocks soaked in distilled water and a NaCl solution were used for the second set, except that the sensors were placed on opposite surfaces but on the same concentric centre point, as shown in Figure 5b. Sharma et al. [47] conducted a comparison study on sodium (Na), potassium (K) and magnesium (Mg), and it was concluded that NaCl solutions are more corrosive than the chloride of Mg and K.

## 5. Results and Discussion

In this Section, the recorded results, with the locations of the maximum and minimum outputs critical for determining corrosion, are tabulated and analysed. 

### 5.1. Coplanar-Placed Sensors

Three pulses were taken from each wave signal obtained from each block and then separated into three Sections, one for each pulse. In Table 1, the rows are divided into three Sections for two different cases (i.e., configurations of sensors placed coplanarly and on opposite surfaces), with each representing a position on the wave signal obtained, as illustrated in Figure 6; Figure 7 shows different cases.

The first Section ranges from 0 to 53,333 × 10^−4^ s, the second from 53,334 × 10^−4^ to 10,666 × 10^−4^ and the third from 10,667 × 10^−4^ to 160,000 × 10^−4^, with the maxima and minima obtained within these ranges shown in Figure 6 and Figure 7. The values in Table 2 are the times at which these maximum and minimum voltage signals occurred within the three Sections (indicated as sample Section 1, Section 2 and Section 3 in Figure 6 and Figure 7). The time slots in which the maximum and minimum occurred in one Section were subtracted from those in the others, e.g., the timeslot with the maximum value in Section 2 was subtracted from that in Section 1 and that in Section 3 subtracted from that in Section 2. By doing this, we obtained offsets for the maximum and maximum values. Then, by averaging the two results, we arrived at the average time differences (as shown in Table 3) for both cases among the three different blocks. 

From the results, it is evident that the concept of using EMI to detect the presence of corrosive chemicals is possible. The signals obtained from the different concrete specimens were different, as they were subjected to different environmental conditions, and research could results in the development of novel techniques to detect salts and predict corrosion in concrete. However, further research is needed in this area as this research was a proof of concept. Further research on different salts types, salt concentration, period of exposure to salts and the mortar ratio would further enhance the research methodology.

### 5.2. Coil Placed on Opposite Surfaces

For the coplanar-placed sensors, the difference in the time delays between the pulses for the dry concrete block and the concrete block soaked in distilled water was approximately 0.25 s (50,392.5 × 10^−4^ to 47,934.5 × 10^−4^ s from Table 2), with the latter showing less delay than the former, while the block soaked in the NaCl solution produced a higher difference than both. The differences were 1.5 s between the dry block and that submerged in NaCl, and 1.8 s between the blocks submerged in distilled water and NaCl. Therefore, the time delay (time delay between the transmitting and receiving signal) in a concrete block soaked in a NaCl solution was more easily detectable than those in a dry block and one exposed to water. Although the same conclusion can be drawn for the sensor coils placed on opposite surfaces, the time delays were much larger, as shown in Table 3.

Comparing the two configurations, it is clear that the sensors placed coplanarly produced much stronger signals, (i.e., lower average time delays) than those placed on opposite surfaces. This was due to a decreased distance between the transmitter and receiver, which resulted in a larger EMI strength that satisfied Equation (4). It should be noted that the potential responses shown are as measured and they are strictly limited to qualitative interpretations. Further research with extended samples is currently being undertaken for a quantitative understanding on the origin of these signals.

## 6. Conclusions

A sensor was developed based on EMI principles following Faraday’s law of EMI. The MLC design was successful to the extent that it was capable of displaying the differences between a dry concrete block and blocks submerged in distilled water and a NaCl solution. The differences observed among the three blocks were in the time delays between the pulses extracted from the data acquisition process, which, it can be stated, were due to the blocks’ different moisture and chemical contents, which changed their resistivity levels. Therefore, if a steel-reinforced concrete structure was suspected of containing any corrosive chemicals, (i.e., NaCl), this chemical could be identified by the proposed sensor technology. The newly developed prototype proves the concept of detecting corrosive chemicals present in concrete through the use of EMI-based technology. This was evident, as the results depicted that the presence of salts alters the received signal. The time delay for the dry block, one submerged in distilled water and one submerged in NaCl are 0.45, 0.25 and 1.80 s, respectively, for the prototype.

Interesting topics for further research on the time delay between the EMI-induced pulses detected in each concrete block could be using various thicknesses for testing and acquiring signals from sensors placed on both coplanar and opposite surfaces; constructing coils with different numbers of turns, sizes and wire materials to identify the parameters for an optimum sensor design specification; and determining benchmark values for the time delay of other corrosive solutions (e.g., MgCl_2_, CaCl_2_, and BaCl_2_).

## 7. Patents

This research project is part of the patent work in the field of monitoring corrosion in reinforced steel. An EMI-based sensor for non-destructive corrosion estimation in concrete has been patented under IP Australia, 2,018,100,760, 2018.

## Figures and Tables

**Figure 1 sensors-19-01959-f001:**
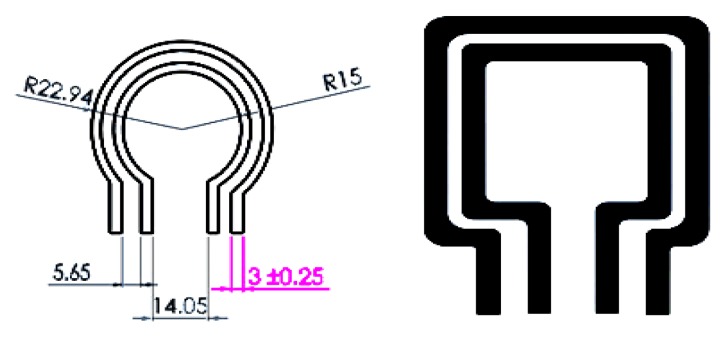
Two different configurations of design of the single-loop coil (SLC) sensor probe: Mild steel (circular shape) and brass (rectangular shape).

**Figure 2 sensors-19-01959-f002:**
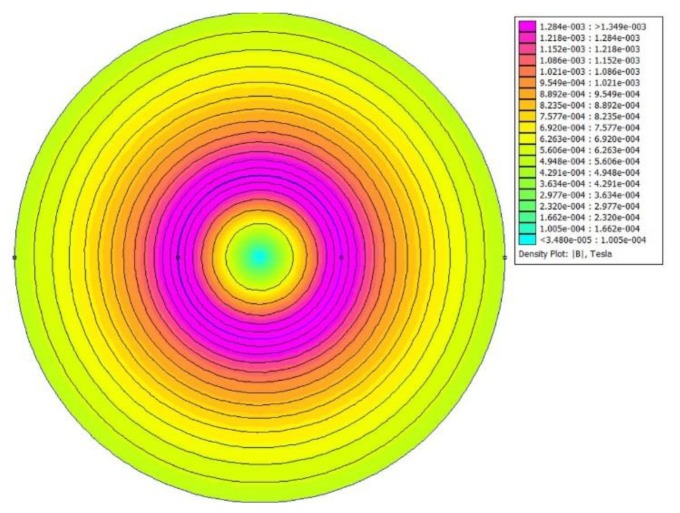
Simulation of the flux distribution under an electromagnetic induction (EMI) sensor.

**Figure 3 sensors-19-01959-f003:**
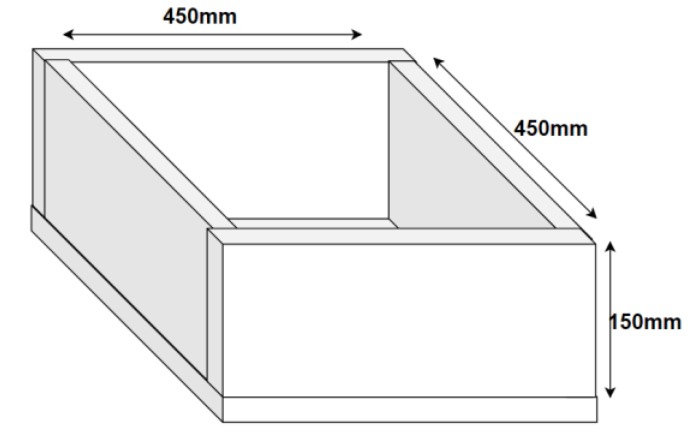
Schematic of complete box frame with inner dimension of 150 mm × 450 mm × 450 mm.

**Figure 4 sensors-19-01959-f004:**
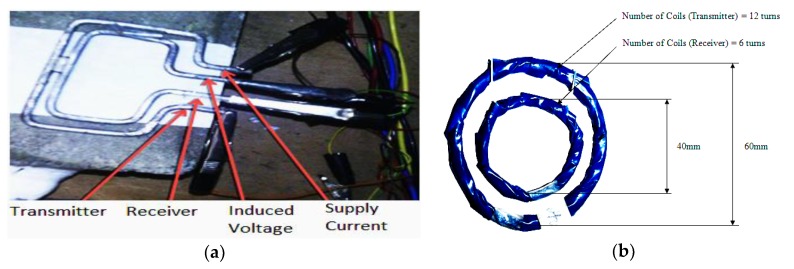
(**a**) Mild steel SLC setup and (**b**) multiple-loop coil (MLC) configuration made from Category 5 (CAT 5) cable.

**Figure 5 sensors-19-01959-f005:**
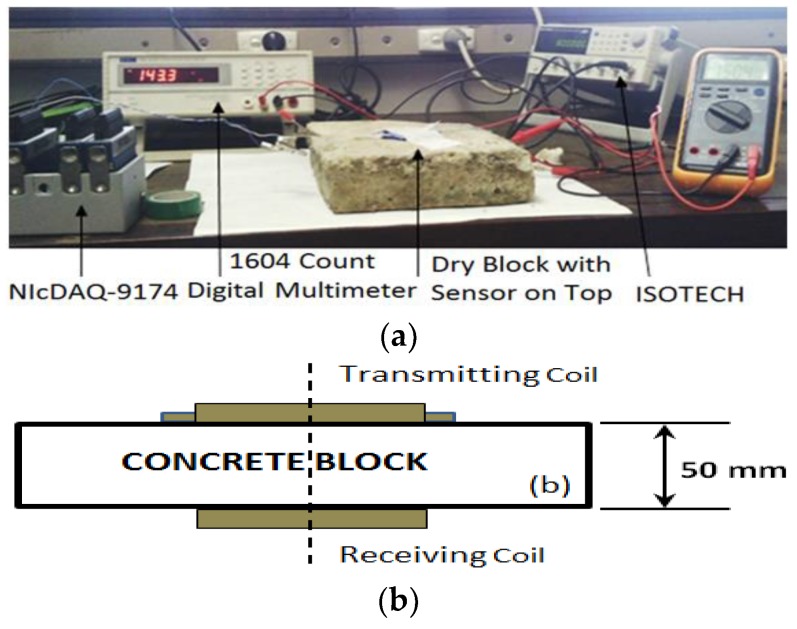
(**a**) MLC sensor test setup and (**b**) transmitter and receiver placed on opposite surfaces of block but lying on same centre line.

**Figure 6 sensors-19-01959-f006:**
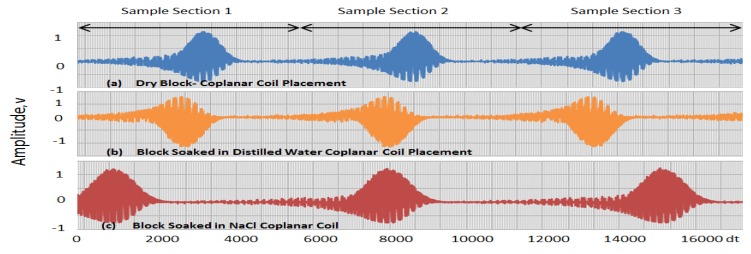
Graphs of signals (voltages) obtained from tests conducted with coplanar-placed sensors on (**a**) dry block, (**b**) concrete block submerged in distilled water and (**c**) concrete block submerged in NaCl solution.

**Figure 7 sensors-19-01959-f007:**
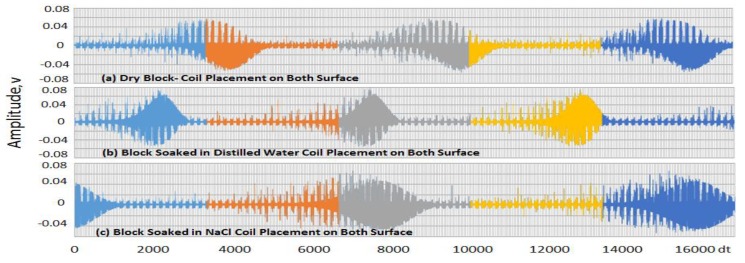
Voltages obtained from tests conducted with coils placed on opposite sides on (**a**) dry concrete block, (**b**) concrete block soaked in distilled water and (**c**) concrete block soaked in NaCl solution.

**Table 1 sensors-19-01959-t001:** Maximum and minimum values of voltage (V) signals obtained for/from three concrete blocks with sensors placed coplanarly (COP) and on opposite surface (OPP).

No.	Time Interval (s)		Dry Block (V)		Distilled Water (V)		NaCl (V)	
			COP	OPP	COP	OPP	COP	OPP
1	0–53,333	Min	−0.778	−0.053	−1.138	−0.053	−0.761	−0.044
		Max	1.147	0.058	0.793	0.073	1.213	0.044
		Pk–Pk	1.925	0.111	1.931	0.126	1.974	0.088
2	53,334–10,666	Min	−0.767	−0.057	−1.14	−0.054	−0.758	−0.052
		Max	1.14	0.058	0.795	0.08	1.228	0.066
		Pk–Pk	1.907	0.115	1.935	0.134	1.986	0.118
3	10,667–16,000	Min	−0.774	−0.056	−1.128	−0.053	−0.753	−0.053
		Max	1.14	0.059	0.785	0.074	1.247	0.066
		Pk–Pk	1.914	0.115	1.913	0.127	2	0.119
		Pk–Pk Average	1.915	0.113	1.926	0.129	1.986	0.1185

**Table 2 sensors-19-01959-t002:** Locations of maximum and minimum voltage signals for the identification of time delays for sensors placed coplanarly (COP) and on opposite surface (OPP).

Time (10^−4^ s)	Dry Block (s × 10^−4^)		Distilled Water (s × 10^−4^)		NaCl (s × 10^−4^)	
	cop	opp	cop	opp	cop	opp
Min 1	30,086	36,601	25,913	18,458	8569	32
Max 1	30,346	32,165	24,565	20,166	8741	324
Min 2	80,026	94,077	73,813	68,790	74,469	69,924
Max 2	81,198	86,201	72,425	71,586	74,329	67,376
Min 3	130,939	148,118	121,758	119,935	139,934	140,901
Max 3	131,063	142,722	120,458	122,555	139,618	143,985
Max2 − Max1	50,852	54,036	47,860	51,420	65,588	76,611
Max3 − Max1	49,865	56,521	48,033	50,969	65,289	76,609
Max (Diff)	50,358	55,278	47,946	51,194	65,438	76,610
Min2 − Min1	49,940	57,476	47,900	50,332	65,900	70,975
Min3 − Min2	50,913	54,041	47,945	51,145	65,465	70,977
Min (Diff)	50,426	55,758	47,922	50,738	65,682	70,976
Average Time Difference	50,392	55,518	47,934	50,966	65,560	73,793

**Table 3 sensors-19-01959-t003:** Time difference for both configurations (coplanar and opposite).

**Samples**	**Dry Block**	**Distilled Water**	**NaCl**	**Opposite**
Dry Block	-	0.45 s	1.83 s	
Distilled Water	0.25 s	-	2.20 s	
NaCl	1.80 s	1.50 s	-	
**Coplanar**				

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
