# Peer review of "A Prototype of an Electromagnetic Induction Sensor for Non-Destructive Estimation of the Presence of Corrosive Chemicals Ensuing Concrete Corrosion"

_sensors, 2019, doi:10.3390/s19091959_

Round 1
Reviewer 1 Report
I believe this work would be of interest to the readership of Sensors. However, some aspects still need to be clarified before considering publication.
1. Researches on microwave antenna sensor for Structure Health Monitoring should be discussed:
(1) J. Yao, S. Tjuatja, and H. Huang, “Real-time vibratory strain sensing using passive wireless antenna sensor,” IEEE Sensors J., vol. 15, no. 8, pp. 4338–4345, Aug. 2015.(2) H. Cheng, S. Ebadi, X. Ren, and X. Gong, “Wireless passive high temperature sensor based on multifunctional reflective patch antenna up to 1050 degrees centigrade,” Sens. Actuators A, Phys., vol. 222, pp. 204–211, Feb. 2015.(3) D. Girbau, A. Ramos, A. Lazaro, S. Rima, and R. Villarino, “Passive wireless temperature sensor based on time-coded UWB chipless RFID tags,” IEEE Trans. Microwave Theory Tech., vol. 60, no. 11, pp. 3623–3632, Nov. 2012.(4) J. W. Sanders, J. Yao, and H. Huang, “Microstrip patch antenna temperature sensor,” IEEE Sensors J., vol. 15, no. 9, pp. 5312–5319, Sep. 2015.
2. EM simulation should be added to evaluate the sensor design concept.
3. If the sensor was placed coplanar on the concrete block’s top surface, what is the mutual coupling between the transmitter and receiver?
Author Response
I believe this work would be of interest to the readership of Sensors. However, some aspects still need to be clarified before considering publication.
1. Researches on microwave antenna sensor for Structure Health Monitoring should be discussed:
(1) J. Yao, S. Tjuatja, and H. Huang, “Real-time vibratory strain sensing using passive wireless antenna sensor,” IEEE Sensors J., vol. 15, no. 8, pp. 4338–4345, Aug. 2015.(2) H. Cheng, S. Ebadi, X. Ren, and X. Gong, “Wireless passive high temperature sensor based on multifunctional reflective patch antenna up to 1050 degrees centigrade,” Sens. Actuators A, Phys., vol. 222, pp. 204–211, Feb. 2015.(3) D. Girbau, A. Ramos, A. Lazaro, S. Rima, and R. Villarino, “Passive wireless temperature sensor based on time-coded UWB chipless RFID tags,” IEEE Trans. Microwave Theory Tech., vol. 60, no. 11, pp. 3623–3632, Nov. 2012.(4) J. W. Sanders, J. Yao, and H. Huang, “Microstrip patch antenna temperature sensor,” IEEE Sensors J., vol. 15, no. 9, pp. 5312–5319, Sep. 2015.
All suggested papers were taken into consideration and it has been referenced from 42-45 in line 89
2. EM simulation should be added to evaluate the sensor design concept.
EM simulation are added in the paper under 3.Sensor Design (lines 157-176)
3. If the sensor was placed coplanar on the concrete block’s top surface, what is the mutual coupling between the transmitter and receiver?
The sensor was placed coplanar it is inductively coupled. (The transmitter and receiver search coil was placed in concentric alignment coplanarly with 10mm air gab, the sensor act as an inductively coupled (magnetic fields fluctuate around the transmitter coil and as a result the secondary coil begins to induce current).
Reviewer 2 Report
The paper proposed two sensors that can recognize the presence of chemical contents (i.e NaCl) inside concrete structures. The design of sensors is based on Single-loop Coil (SLC) and Multiple-loop Coil (MLC) methods. The authors concluded that the sensors developed can be applied to detect the concrete corrosion as a NDE technique. Although the work is an interesting topic, unfortunately, it does not reach to be published in a journal with its present form. Major revision needs to be performed.
I strongly suggest to revise the manuscript and to answer the comments/questions as listed below. I also would be happy to read the article after revision to make sure that the authors had addressed all reviewer’s comments and suggestions.
Equation 1: Variables in Eq.1 need to be explained in the manuscript.
Lines 138-140: The author stated that “To obtain greater accuracy, the lengths and thicknesses of the two coils, the gaps between the receiver and transmitter, and other factors, such as the shapes and material variations of the coils, were taken into account”. How did you consider the effects of these factors? Where is it presented in the manuscript?
Section 4:
The dimension of concrete specimens needs to be added fully in the manuscript.
The explanation in the manuscript is not clear, so the authors need to check it again. The results in Table 1, Table 2, Figure 5 and Figure 6 should be explained in more detail.
Lines 163-164: Why did the authors select the time duration to soak the blocks of 24 hours? How are the results affected by the length of the soaking period? The third block was left dry but how did you determine the degree of its dryness?
Table 1: It shows “the Maximum and Minimum Values of Voltage (V) Signals Obtained For/Form Three Concrete Blocks with Sensors Placed Co-Planarly (COP) and On Opposite Surface (OPP)”. What are your purposes by showing these values? Some explanation needs to be added in the manuscript.
What is the meaning of determination of the time delay and time difference? It needs to be added in the manuscript.
Lines 239-241: The authors stated that “This research proved that the proposed concept for the design of a sensor for concrete in different environments based on EMI principles was more effective than other NDE techniques presented in various research publications [34-38]”. Authors need to clarify: What is more effective than other research?
I think at least one of two following requirements should be considered and added to the manuscript. If they can not be added, I strongly recommend that an explanation needs to be given to clarify clearly the effects of factors below.
The soaking period should be changed and the obtained results will be compared.
The molar mass of NaCl solution should be varied and the obtained results will be compared.
Abstract: The authors stated that “As this new sensor was able to detect and differentiate among the chemical contents present in a concrete structure, it could be used as a NDE technique for the detection and non-destructive evaluation of concrete corrosion”. How can you conclude that the structure is corroded, especially new structures located near the sea? Because the NaCl can occur in the concrete, but the structure might be not corroded in initial time.
The title: As described in Comment 9, I think it is better if the title will be changed as “EMI Sensor for Non-destructive Estimation of the Presence of Corrosive Chemicals in Concrete”. Therefore, the authors need to check again the whole of the manuscript, especially the abstract and conclusion sections.
If you apply the proposed sensors for real inspection, how do you come to the conclusion that the corrosive chemical inside concrete is NaCl?
Author Response
The paper proposed two sensors that can recognize the presence of chemical contents (i.e NaCl) inside concrete structures. The design of sensors is based on Single-loop Coil (SLC) and Multiple-loop Coil (MLC) methods. The authors concluded that the sensors developed can be applied to detect the concrete corrosion as a NDE technique. Although the work is an interesting topic, unfortunately, it does not reach to be published in a journal with its present form. Major revision needs to be performed.
I strongly suggest to revise the manuscript and to answer the comments/questions as listed below. I also would be happy to read the article after revision to make sure that the authors had addressed all reviewer’s comments and suggestions.
Equation 1: Variables in Eq.1 need to be explained in the manuscript.
The explanation are been added in lines 96-97
Lines 138-140: The author stated that “To obtain greater accuracy, the lengths and thicknesses of the two coils, the gaps between the receiver and transmitter, and other factors, such as the shapes and material variations of the coils, were taken into account”. How did you consider the effects of these factors? Where is it presented in the manuscript?
The explanation are been added in lines 142-149
Section 4:
The dimension of concrete specimens needs to be added fully in the manuscript.
The dimension of concrete specimens are been added in lines 192-202.
The explanation in the manuscript is not clear, so the authors need to check it again. The results in Table 1, Table 2, Figure 5 and Figure 6 should be explained in more detail.
The explanation are been added in lines 251-257
Lines 163-164: Why did the authors select the time duration to soak the blocks of 24 hours? How are the results affected by the length of the soaking period? The third block was left dry but how did you determine the degree of its dryness?
The time period for soaking period was chosen as the concrete blocks were saturated within the 24 hour period. This was evident as concrete blocks were weighed after 12, 24, 36 and 48hours and it was observed that the weight did not change after 24 hour period. This is shown in lines 188-191.
Table 1: It shows “the Maximum and Minimum Values of Voltage (V) Signals Obtained For/Form Three Concrete Blocks with Sensors Placed Co-Planarly (COP) and On Opposite Surface (OPP)”. What are your purposes by showing these values? Some explanation needs to be added in the manuscript.
The purpose is explained in lines 251-257
What is the meaning of determination of the time delay and time difference? It needs to be added in the manuscript.
Both were written for same meaning, but change has been made “Delay” for consistence in meaning.
Lines 239-241: The authors stated that “This research proved that the proposed concept for the design of a sensor for concrete in different environments based on EMI principles was more effective than other NDE techniques presented in various research publications [34-38]”. Authors need to clarify: What is more effective than other research?
The statement has been revised in lines 291-294
I think at least one of two following requirements should be considered and added to the manuscript. If they cannot be added, I strongly recommend that an explanation needs to be given to clarify clearly the effects of factors below.
The soaking period should be changed and the obtained results will be compared.
The explanation are been added in lines 188-191
The molar mass of NaCl solution should be varied and the obtained results will be compared.
This can be done as further research but a similar work has been done and explained in lines 228-230
Abstract: The authors stated that “As this new sensor was able to detect and differentiate among the chemical contents present in a concrete structure, it could be used as a NDE technique for the detection and non-destructive evaluation of concrete corrosion”. How can you conclude that the structure is corroded, especially new structures located near the sea? Because the NaCl can occur in the concrete, but the structure might be not corroded in initial time.
The statement has been revised in lines 14-25
The title: As described in Comment 9, I think it is better if the title will be changed as “EMI Sensor for Non-destructive Estimation of the Presence of Corrosive Chemicals in Concrete”. Therefore, the authors need to check again the whole of the manuscript, especially the abstract and conclusion sections.
Title has been revised
If you apply the proposed sensors for real inspection, how do you come to the conclusion that the corrosive chemical inside concrete is NaCl?
This could be possible further research and experiments.
Reviewer 3 Report
The submitted manuscript with ID: sensors-470688 and title: “EMI Sensor for Non-destructive Estimation of Corrosion in Concrete” is an original and interesting contribution on the Non-destructive Evaluation (NDE) techniques for the detection of steel reinforcement corrosion in Reinforced Concrete (RC) structural members. The paper presents the development of two sensors based on the Electromagnetic Induction (EMI) principles. Although the topic falls within the scope of the “Sensors” Journal and it is still open to question since the existing work in this field of study is rather limited, the manuscript has some serious flaws and the overall quality, including value of contribution, methodology and presentation style is rather low for a typical journal paper. The following issues are some sample weaknesses:
- The literature review provided seems shallow, lacking coherence and failing to establish the relevance of the work reported in the paper.
- EMI Sensor Principles are very briefly presented. Equations and variables require further explanations since in these equations any physical meaning of several factors has been eliminated.
- Section “Sensor design” is also very short to comprehend the aspects and the details of the developed sensors.
- Tests are very limited to establish the effectiveness of the presented NDE method. The Authors didn’t succeed in their goal to connect the experimental findings with the developed sensors. This is actually the main drawback of the paper. Further, the relationship between the text and the tables/ figures of sections “Experimental Tests” and “Results and Discussion” is very weak; the text leads the readers to believe that the tables/ figures will provide the desired information and/or clarification of the work done but the tables/ figures do not provide this. Discussion and further commentary of the tables/ figures should be provided.
- Conclusions seem rather confusing and weak since they are not supported sufficiently by the results. More specific and sound concluding remarks should be reported concering the findings of this work.
On the basis of the above comments, the paper could be suggested to be rejected. However, it is recommended the manuscript to be re-submitted after extensive revision, providing the improvements suggested.
Author Response
The submitted manuscript with ID: sensors-470688 and title: “EMI Sensor for Non-destructive Estimation of Corrosion in Concrete” is an original and interesting contribution on the Non-destructive Evaluation (NDE) techniques for the detection of steel reinforcement corrosion in Reinforced Concrete (RC) structural members. The paper presents the development of two sensors based on the Electromagnetic Induction (EMI) principles. Although the topic falls within the scope of the “Sensors” Journal and it is still open to question since the existing work in this field of study is rather limited, the manuscript has some serious flaws and the overall quality, including value of contribution, methodology and presentation style is rather low for a typical journal paper. The following issues are some sample weaknesses:
- The literature review provided seems shallow, lacking coherence and failing to establish the relevance of the work reported in the paper.
- EMI Sensor Principles are very briefly presented. Equations and variables require further explanations since in these equations any physical meaning of several factors has been eliminated.
EMI is well-known principle, so author did not mention much about this in the paper.
- Section “Sensor design” is also very short to comprehend the aspects and the details of the developed sensors.
More details are added in lines 128-179.
- Tests are very limited to establish the effectiveness of the presented NDE method. The Authors didn’t succeed in their goal to connect the experimental findings with the developed sensors. This is actually the main drawback of the paper. Further, the relationship between the text and the tables/ figures of sections “Experimental Tests” and “Results and Discussion” is very weak; the text leads the readers to believe that the tables/ figures will provide the desired information and/or clarification of the work done but the tables/ figures do not provide this. Discussion and further commentary of the tables/ figures should be provided.
Revision have been made respective sections.
- Conclusions seem rather confusing and weak since they are not supported sufficiently by the results. More specific and sound concluding remarks should be reported concering the findings of this work.
More explanation has been added in conclusion.
On the basis of the above comments, the paper could be suggested to be rejected. However, it is recommended the manuscript to be re-submitted after extensive revision, providing the improvements suggested.
Round 2
Reviewer 1 Report
All concerns are met.
Reviewer 2 Report
I found that the authors have adequately revised the paper according to all my comments and their answers are quite acceptable. I think the revised manuscript can be published in the Journal.
Reviewer 3 Report
The article is well-revised and ameliorated based on the recommendations of the first review. Hence it can be published without further re-review.